Slumber in a cell: honeycomb used by honey bees for food, brood, heating… and sleeping

Klein Barrett A. 1 barrett@pupating.org
http://orcid.org/0000-0001-6038-0605 Busby M. Kathryn 2
1 Biology Department, University of Wisconsin-La Crosse , La Crosse, WI , USA
2 Graduate Interdisciplinary Program in Entomology and Insect Science, University of Arizona , Tucson, AZ , USA
Gillespie Joseph
Electronic publication date: 2020 Aug 5
Publication date: 2020
Volume: 8
Electronic Location ID: e9583
Received 2020 Mar 17; Accepted 2020 Jun 30
Copyright: © 2020 Klein and Busby
Copyright year: 2020
Copyright holder: Klein and Busby
License: This is an open access article distributed under the terms of the Creative Commons Attribution License, which permits unrestricted use, distribution, reproduction and adaptation in any medium and for any purpose provided that it is properly attributed. For attribution, the original author(s), title, publication source (PeerJ) and either DOI or URL of the article must be cited.
License URL: https://creativecommons.org/licenses/by/4.0/

Keywords: Sleep, Honeycomb, Heating bees, Thermography, Discontinuous ventilation

Funding: Deutscher Akademischer Austausch Dienst DAAD #A0670415 This work was supported by the Deutscher Akademischer Austausch Dienst (DAAD #A0670415). The funders had no role in study design, data collection and analysis, decision to publish, or preparation of the manuscript.

==============================
Sleep appears to play an important role in the lives of honey bees, but to understand how and why, it is essential to accurately identify sleep, and to know when and where it occurs. Viewing normally obscured honey bees in their nests would be necessary to calculate the total quantity and quality of sleep and sleep’s relevance to the health and dynamics of a honey bee and its colony. Western honey bees (Apis mellifera) spend much of their time inside cells, and are visible only by the tips of their abdomens when viewed through the walls of an observation hive, or on frames pulled from a typical beehive. Prior studies have suggested that honey bees spend some of their time inside cells resting or sleeping, with ventilatory movements of the abdomen serving as a telltale sign distinguishing sleep from other behaviors. Bouts of abdominal pulses broken by extended pauses (discontinuous ventilation) in an otherwise relatively immobile bee appears to indicate sleep. Can viewing the tips of abdomens consistently and predictably indicate what is happening with the rest of a bee’s body when inserted deep inside a honeycomb cell? To distinguish a sleeping bee from a bee maintaining cells, eating, or heating developing brood, we used a miniature observation hive with slices of honeycomb turned in cross-section, and filmed the exposed cells with an infrared-sensitive video camera and a thermal camera. Thermal imaging helped us identify heating bees, but simply observing ventilatory movements, as well as larger motions of the posterior tip of a bee’s abdomen was sufficient to noninvasively and predictably distinguish heating and sleeping inside comb cells. Neither behavior is associated with large motions of the abdomen, but heating demands continuous (vs. discontinuous) ventilatory pulsing. Among the four behaviors observed inside cells, sleeping constituted 16.9% of observations. Accuracy of identifying sleep when restricted to viewing only the tip of an abdomen was 86.6%, and heating was 73.0%. Monitoring abdominal movements of honey bees offers anyone with a view of honeycomb the ability to more fully monitor when and where behaviors of interest are exhibited in a bustling nest.

Introduction

Sleep is a behavior steeped in mystery, yet it appears to offer essential benefits (Rattenborg et al., 2007; Cirelli & Tononi, 2008). Sleep may specifically assist with honey bee communication (Klein et al., 2010, 2018) and memory (Zwaka et al., 2015), so accurately identifying sleep and knowing when and where it occurs is essential for further investigating sleep’s role in honey bee ecology. To better understand sleep’s benefits, or the detriments that come with sleep loss, it is essential to monitor sleep, including when it occurs in dark, hidden places. Several species of honey bees (Apis spp.) nest inside cavities, and all species of honey bees spend periods of their lives concealed inside honeycomb cells, within which much of the colony’s behaviors occur and without which the colony would inevitably perish. Honeycomb is where honey bees store honey and pollen, rear brood, and where some appear to sleep. Young adult workers (callows/cell cleaners) appear to sleep almost exclusively inside cells (Klein et al., 2008, 2014), but workers spend less and less time sleeping inside cells as they age and change tasks, with foragers spending none of their time asleep inside cells (Kaiser, 1988; Klein et al., 2008). If significant periods of sleep occur within honeycomb cells, it would be wise to take inside-cell behavior into account.

Looking for sleep within the dark confines of a honey bee nest requires close examination or monitoring of bees, and modifying a nest to expose the inner workings of a colony has a long and curious history (Crane, 1983), including displaying comb by adding transparent glass jars to hives (Kritsky, 2010), and designing research-friendly observation hives (Von Frisch, 1967; Seeley, 1995). Observation hives increase visibility by encapsulating frames of honeycomb between panes of glass, and this innovation has led to revolutionary discoveries in animal behavior (Von Frisch, 1967; Seeley, 2019). Lindauer (1952) cleverly and patiently recorded the behaviors of individual honey bee workers in a modified observation hive that allowed him to observe activities within a subset of honeycomb cells. Lindauer made a point of recording when a bee was “Müssig” (idle), and concluded that time spent exhibiting this behavior, a portion of which occurred inside cells, far outweighed the time spent performing other tasks (e.g., bee #107: 68 h 53 min out of the total 176 h 45 min observed). He referred to bees seeking out undisturbed resting places, like empty or egg-containing cells, and spending long, calm periods inside these cells. Lindauer used an icon of a couch or bed to symbolize this behavior, despite including cleaning movements and other somewhat superficially immobile states in his calculations. We now know that a portion of this relatively immobile time in cells is devoted to heating brood in adjacent cells (Kleinhenz et al., 2003), but to what extent the remaining time is spent sleeping has never been rigorously established.

Few studies following Lindauer’s observations have addressed immobility or potentially sleep-like states in cells. These include reports of honey bees exhibiting a resting state (“Ruhezustand”, Sakagami, 1953), a “motionless” state (Moore et al., 1998), “rest” (Kaiser, 1988), or ventilatory signs indicative of sleep (Sauer, Menna-Barreto & Kaiser, 1998; Kleinhenz et al., 2003). Sleep is typically defined by several behavioral criteria, most important of which is an increased threshold of responsivity to a stimulus. An animal with an increased response threshold exhibits a specific posture during states of relative immobility that are reversible (Flanigan, Wilcox & Rechtschaffen, 1973). This suite of sleep signs is internally controlled (Tobler, 1985), meaning that if deprived of the state, the organism will respond with an increased expression of the behavior. Kaiser (1988) and Sauer, Herrmann & Kaiser (2004) confirmed that these coincident behavioral traits exist in honey bees.

Not all sleep signs can be observed simultaneously, so a dependable indicator of sleep would be of great value. Antennal immobility is a feature that has been used as a proxy for sleep (Eban-Rothschild & Bloch, 2008; Hussaini et al., 2009; Zwaka et al., 2015; Vázquez et al., 2020) because the amount of antennal immobility per unit time correlates with an increased response threshold (Kaiser, 1988). Another feature that covaries with antennal immobility (Sauer et al., 2003) and could, therefore, be an alternative proxy for sleep, is discontinuous ventilation. The honey bee’s metasoma (hereafter referred to as “abdomen”) moves in anterior-posterior pumping motions (pulses) at various rates and degrees of continuity. Easily observed extremes include “continuous” and “discontinuous” ventilation, in which the interim between anterior-posterior abdominal motions is consistently brief (continuous) or occasionally broken by extended pauses of at least 10 s (discontinuous; Kleinhenz et al., 2003). Honey bees exhibit continuous and discontinuous ventilation inside cells (Kleinhenz et al., 2003) and outside cells (Kaiser, 1988), suggesting higher and lower rates of respiration (Bailey, 1954). A discontinuously ventilating honey bee appears to almost always have a higher response threshold, a hallmark of sleep (B. Klein, 2020, in preparation). Because it can be difficult or impossible to gauge antennal movement in a nest, especially if a bee is inserted in a honeycomb cell, ventilatory activity holds promise as a more suitable indicator of sleep under natural or close-to-natural conditions. It is worth noting that antennal immobility may even be a misleading indicator of sleep because brood-incubating (heater) bees, which may appear to be asleep on the comb surface because of an absence of large body movements, also exhibit slow to no antennal movement (Bujok et al., 2002).

Kleinhenz et al. (2003) modified Lindauer’s (1952) hive manipulation and used ventilatory rates, in part, to distinguish resting versus heating honey bees. We adopted this approach to peer at the undisturbed activities of worker bees in comb cells to see if what can be seen outside a cell (tip of abdomen; Fig. 1A) can serve as a reliable indication of what is going on with the rest of a bee’s body inside the cell (Figs. 1B–1D). We hypothesized that ventilatory rate (continuous vs. discontinuous ventilation) and the presence/absence of larger movements of the abdomen can be used to predict behavior of bees inside cells. If correlations are robust between behavior of honey bees inside cells with behavior that is observable outside cells, someone observing only the posterior tips of honey bees when bees are inserted in honeycomb should be able to identify the bees’ behaviors, including sleep.

Figure 1 Visibility of honey bee workers deep inside cells in observation hives.

(A) Clear view of posterior end of worker abdomen, center. This is the typical view of an adult bee inside a cell when removing frames from a beehive, or, in this case, when viewed through the glass window of an observation hive (Würzburg, Germany, 2006). (B–D) Unusually clear views of bees sleeping inside cells built on windows of an observation hive (USA, 2019). Sometimes bees construct comb against the glass of an observation hive, exposing the occasional bee’s activities within comb cells. Photos by Barrett Klein.

Materials and Methods

We set up a small colony of honey bees in an observation hive with honeycomb positioned so that the interiors of some cells were visible. We recorded bees’ behaviors inside the visible cells using an infrared-sensitive camera and a thermographic camera, first by surveying all of the visible cells, then by zooming into and recording examples of behaviors for later analysis. We classified behaviors into four categories based on body movement, ventilatory rate and surface temperatures. To test the viability of identifying behaviors based solely on viewing the portion of a bee that is visible when honeycomb is exposed in a more conventional hive, we asked naïve viewers to identify behaviors from a subset of the videos. The viewers used the same four behavioral categories, but the videos were modified so only the posteriors of the bees were visible. Their identifications, made under limited-visibility conditions, were compared to our identifications, which benefitted from careful examination of behavior visible only inside the cells, and surface temperatures visible using the thermographic camera.

Study organisms and hive

We collected one queen, two frames of honeycomb, and 800–1,000 Carniolan worker honey bees (A. mellifera carnica Pollman, 1879) from a bee yard hive, with permission from Dr. Jürgen Tautz and the University of Würzburg (Würzburg, Germany; 49°46′47″ N, 9°58′31″ E). We cut out three sections of comb to fit within a honey bee mating cage (Begattungskästchen; see Kleinhenz et al., 2003) and cleaned out most of the cells along the edges to increase our likelihood of viewing visiting workers. The interiors of 93 empty cells and 22 cells with food were visible along the edges of the hive (Fig. 2). The sections of comb included brood cells, pollen (at least 10 cells), uncapped honey and empty cells. The comb slice on the left side of the hive contained only uncapped honey and empty cells. The middle slice contained 50 capped brood on the left side (nine were one-cell deep from the empty edge cells) and 27 capped brood on the right side (two were one-cell deep). The right slice contained 40 capped brood on the left side (two were one-cell deep) and 41 capped brood on the right side (two were edge cells and five were one-cell deep, with at least six uncapped larval cells toward the back of the comb). Twenty-one hours after inserting the queen, followed by the workers, we introduced 49 uniquely paint-marked callows using nontoxic, oil-based markers (Sharpie, Oak Brook, IL, USA). Marking did not noticeably affect temperature readings in preliminary tests. The intent of introducing individually-marked callows was to increase our likelihood of observing sleep inside cells, because young adults appear to sleep more inside cells than older adults (Klein et al., 2008, 2014). The callows had been incubated at 36 °C and collected within 24 h of emergence, marked on dorsal side of mesosoma (hereafter referred to as “thorax”) and abdomen, and placed in a small cage on top of the new hive, separated from the hive by a screen. After 5 h of callows being exposed to nest odor, the screen was removed and newly marked bees were accepted without any sign of aggression. Ultimately, only a subset of the data recorded came from these introduced, marked workers (see ventilatory and thermal methods, below).

Figure 2 Observation hive with slices of honeycomb and exposed cells.

(A) Still image from infrared video of inhabited hive, with hive entrance leading to tubular tunnel at lower left. (B) Slices of comb were held in place by nails inserted through wood frame and comb slices were aligned at natural distances from each other. (C) Comb on right is angled prior to study to show width and some of cell contents, including uncapped and capped brood. Bottom, back corner of each comb was removed to allow easy travel by workers through the hive. Photos by Barrett Klein.

The hive allowed for unrestricted access to the outdoors for the duration of the study (20–24 August 2008, with data collected from 23–24 August) via an entrance tunnel. The hive window was replaced with a sheet of transparent polypropylene giftwrap (pbsfactory, Artikel 00347, Rheinland-Pfalz, Germany) that remained in place for the entire study to allow for thermographic recordings. The ambient temperature of the small room was maintained high enough by using a space heater that insulation was not used to cover the hive during any portion of the short study. Diet was supplemented with honey and sugar water ad libitum.

Behaviors of interest

Four categories of behaviors were recorded: sleeping, maintaining cells, eating, and heating (Table 1). Sleeping was identified by a bee’s discontinuous ventilation and otherwise relative immobility (see description, above). Maintaining cells (i.e., cleaning or building) was identified by occasional large body movements, or obvious mandibular activity while continuously ventilating (although ventilation could be difficult to assess during large body movement episodes) in a cell devoid of food. Sakagami (1953) identified cleaners as externally quiet or irregularly moving, rotating once in a while in a cell. Eating was rarely observed, but obvious when it did occur; a continuously ventilating bee extended her tongue into a cell containing liquid. Heating was identified when a bee with a relatively hot thorax was deep in a cell, continuously ventilating and otherwise immobile. We have no data for bees packing pollen and, because we had no uncapped brood in exposed cells, we have no data involving development or direct tending of brood.

Table 1 Criteria used to define behaviors of honey bees when visible in comb cells, or when visibility was limited to posterior ends of abdomens.

Honey bees were observed inside exposed comb cells (i.e., in cells on edge of comb, visible through plastic window), or visibility was limited by digitally obscuring cell interiors (for testing predictability of behavior from observations of abdominal tip alone; Fig. 4). Ventilatory movements appear as anterior-posterior abdominal pulses, occasionally consolidated into “bouts.” Continuous ventilation = respiratory pulses separated by <10 s of immobility (rarely, if ever, >10 s); discontinuous ventilation = respiratory pulses in bouts, separated by >10 s of immobility (Kleinhenz et al., 2003). Sample size refers to bees for which we collected respiratory rate data.

Behavior	Criteria used to identify behavior when cell interior was visible	Criteria used to identify behavior when cell interior was digitally obscured (test videos)	n (bees)	
Sleeping	Discontinuously ventilating, otherwise relatively immobile (Klein et al., 2008)	Discontinuously ventilating, otherwise immobile	12	
Maintaining cells	Body active in empty cell, often obscuring continuous ventilation; mandibular or antennal movement commonly observed (i.e., cleaning or building cells)	Continuously ventilating, often coupled with larger body movements (in and out, or rotating in cell; Sakagami, 1953)	10	
Eating	Tongue extended in cell containing liquid; continuously ventilating	Continuously ventilating with possible body movement, and only partially in cell (cell contents prevent bee from going deeper)	3	
Heating	Continuously ventilating, otherwise immobile while deep in cell; thorax obviously hotter than surroundings (when viewed using thermal camera)	Continuously ventilating, otherwise immobile	12	

We conducted three sets of analyses: (1) We surveyed behaviors of bees visible inside cells across multiple time points, and after zooming in with the video camera to record exemplars of the different behaviors, we (2) analyzed a subset of the surveyed bees for ventilatory rates, then (3) used the thermography to measure surface temperatures associated with a subset of the bees that had been analyzed for ventilatory rates. By restricting thermal analyses to only those bees for which we acquired ventilatory rate data, we could test whether heating bees could be identified by ventilatory rates (and relative immobility) alone.

For survey data (Dataset S1 and S2), we scanned the cells with visible interiors (n = 115) at 49 discrete time points, and recorded behavior for all bees inside cells. Surveys were separated by at least 10 min, and, because cell maintenance was so commonly observed (considered the default behavior when not explicitly announced by B.A.K.), surveys sometimes started when a behavior other than cell maintenance was detected to ensure sampling of these other behaviors. Each survey involved examining every bee inserted at least partially inside comb cells for at least three to five seconds if obviously maintaining cells (i.e., cleaning or building), or eating, and for longer (>10 s, and sometimes for several minutes) if a worker appeared to be sleeping or heating (Table 1; Fig. 3; Movies S1–S4). Surveys stopped immediately after identifications of behaviors, or after a few minutes of close-up filming for subsequent ventilatory analysis. B.A.K. identified behaviors in real time and identified individually paint-marked bees by briefly shining a tiny white light on the abdomen. Each behavioral count represented a unique bee within each survey, but some bees were undoubtedly repeatedly measured across surveys.

Figure 3 Still images from infrared-sensitive videos of behaviors, with all honey bees head-first inside cells.

(A) Sleeping bee, center cell, is facing left, venter up. (B) Bee is eating, with mouthparts extended and body less fully inserted in cell; facing left, venter down. (C) Heating bee is facing left, venter facing observer (sideways). (D) Sleeping bee, center, is facing left, venter down, and is to be compared with heating bee, at right, facing right, dorsum facing observer (sideways). All other bees inside cells are maintaining (cleaning or building) cells, including two visible in (A) and three in (B), indicated by black circles superimposed on the middle of their thoraces. Contrast and brightness alterations of image serve to highlight sleeping, eating, and heating bees. See Movies S1–S4, from which these images were taken. Images by Barrett Klein.

We collected ventilatory rate data (Dataset S3) from a subset of the surveyed bees. Sleeping and heating bees are relatively immobile (no major head, wing, leg, or body movements), except for ventilatory motions of the abdomen, described above (for more details describing relative immobility during sleep, see Klein et al., 2008). Discontinuous ventilation, identified by bouts of abdominal pulses separated by pauses of stillness exceeding 10 s, occasionally included a single, isolated, apparently spontaneous abdominal jerk during one of these pauses. We excluded a pulse (jerk) if isolated from other pulses by >5 s before and after. We recorded ventilatory rates using JWatcher, an event recorder and analytical software package designed for study of behavior (version 1.0, http://www.jwatcher.ucla.edu/). Recording events with JWatcher entails pressing keys assigned to represent behaviors of interest on a keyboard, with event times automatically recorded. M.K.B. manually pressed one key in time with a pulsing abdomen replayed at 0.3× the normal speed, to increase accuracy and consistency of data transcription. Although we cannot be certain that each behavioral recording represents a unique bee, three steps were taken to increase the likelihood: (1) 12 of the 37 bees were individually marked, and individually-marked bees were analyzed only once; (2) some of the recordings of unmarked bees captured several unmarked bees concurrently, so each was unique within those recordings; and (3) surveys were separated by at least 10 min, and sometimes by several hours.

In addition to behaviors exhibited in exposed cells, we recorded mean surface temperature of a bee’s thorax (Tth) and mean surface temperature of her surroundings (Tsurr) using FLIR’s analysis software package (ResearchIR Max version 4, FLIR Systems, Inc.) from a subset of the bees for which we analyzed ventilatory rates, above (Dataset S4). To calculate the mean temperature of a bee’s thorax (Tth), we drew a circle (within which a mean temperature could be automatically generated) over the region of interest (thorax) in an image taken at the beginning of a thermal recording (several seconds after entering cell), the middle of the recording, and the end (several seconds before exiting cell). To calculate the mean surface temperature of her surroundings (Tsurr) at identical time points, we dragged the same ellipse over three regions bordering the bee’s thorax: above and below thorax, and anterior to head. These regions of interest surrounding the bee’s thorax included almost exclusively cells and cell walls and, unlike a previous study by Klein et al. (2014), did not include any portion of the bee herself. This updated method avoids problems of the bee’s body contributing to the measurement of Tsurr. We report the difference of Tsurr from Tth to indicate the surface temperature of the bee relative to the surface temperature of her surroundings (Tdiff = Tth − Tsurr) (Klein et al., 2014). There was no statistically meaningful difference between using the mean body temperature from the middle time point versus the mean of means across all three time points for any behavior, so we use the middle point when reporting Tth (W = 27, 52, 4, 89; P = 0.65, 0.94, 1.00, 0.07 for bees sleeping, maintaining cells, eating, and heating, respectively) and Tdiff (W = 26, 52, 3, 69; P = 0.58, 0.91, 0.70, 0.61 for bees sleeping, maintaining cells, eating and heating, respectively). Because these bees represent a subset of the bees analyzed above (32 of the 37 bees for which we analyzed ventilatory rates; 11 of the 32 bees were individually marked), the same discussion of unique sampling applies here as well.

To test how predictable a behavior is from observing tips of abdomens alone, we first edited video clips so that they were without sound and a dark gray bar concealed cell contents, revealing only what extended beyond each cell (tip of abdomen and, sometimes, distal portions of hindlegs). We placed a tiny digital mark to indicate the bee(s) of interest in each video (Fig. 4; Movies S5–S8). B.A.K. trained 54 students for 20–30 min by showing and describing behaviors (criteria: Table 1) in 12 video clips (four sleeping, five maintaining cells, two heating, and one showing food in a cell; eating was described but not shown due to lack of additional examples from our recordings). Videos used during training were not used during testing, but were made available to students, had they wished to continue training on their own. Once trained, students independently watched 30 video clips of bees with digitally obscured cell contents—a subset of the 37 bee recordings used in our ventilatory rate analyses (11 sleeping, six maintaining cells, two eating and 11 heating)—and recorded what they believed to be each bee’s behavior.

Figure 4 Still image from infrared test video.

Gray boxes obscure cell interiors, and small light gray rectangle within box on right marks bee of interest. Test yourself with Movies S5–S8, a sample of video clips we used to test the reliability of identifying inside-cell behavior from what is visible outside the cell. (Answers included.) Photo by Barrett Klein.

Recording equipment

We eliminated outdoor light and lit the room with a single desk lamp covered with a red acetate filter (#27 Medium Red, transparency = 4%, peak at 670 nm, Supergel by Rosco, Stamford, CT, USA), selected because honey bees may be less sensitive to frequencies beyond 600 nm (Von Frisch & Lindauer, 1977) or 650 nm (Dustmann & Geffcken, 2000). The same filter was applied to a headlamp, used to facilitate observations. The warm lights were kept away from the hive, and angled to minimize glare that would otherwise affect thermal measures. We filmed under the low, red light, and with an infrared spotlight by using an infrared-sensitive video camera (AGDVC 30, Panasonic, Japan) side-by-side with a thermal camera (FLIR SC660, FLIR Systems Inc., Boston, MA, USA; accuracy 1 °C or 1% of reading, according to FLIR manual and FLIR technical support). We adjusted thermal camera settings to match the emissivity value of a honey bee’s thorax (0.97; Stabentheiner & Schmaranzer, 1987), although wax and other surface temperatures were recorded for Tsurr, and set the transmissivity to that of polypropylene (0.89). The giftwrap used as the observation hive’s window produced a nonlinear error when recording temperature as temperature increased, so we adjusted absolute temperature measurements (Dataset S5; see Klein et al., 2014 for details). Some data were taken using an audio recorder (Olympus VN-4100PC Digital Voice Recorder) and later transcribed. Audio was synchronized with video recordings by the researcher making a noise, followed by announcing the exact time as was recorded on video when the noise was made. Bees were often pointed out when announced, and this served to synchronize thermal imagery with video and audio.

Statistical analysis

Behavior surveys

To determine how prevalent each behavior was within the hive, we compared total counts of bees performing each behavior using a Kruskal–Wallis Rank Sum test. We then conducted post-hoc pairwise tests using six two-sided, non-paired Wilcoxon-Mann-Whitney tests. To avoid multiple testing problems, we corrected resulting P-values using the Holm method in the R function p.adjust(). To account for day/night differences between behaviors, we conducted three Kolmogorov-Smirnov tests using the R function ks.test(). This two-sided test’s null hypothesis states that two sets of data, x and y, were drawn from the same continuous distribution. Therefore, we set our x and y to be the day and night distributions of each behavior, respectively. We performed three such tests to include sleeping, heating, and cell-maintaining behaviors, each time testing that each behavior count distribution remained the same from day to night. We did not perform this test on eating behavior because we did not have a large enough sample. We used local sunrise/sunset times to distinguish day and night (https://www.gaisma.com/en/location/wurzburg.html).

Ventilatory rates

To address whether different behaviors could be distinguished using the time separations between their individual within-bout abdominal ventilation pulses (“pulse separations”), we performed a Kruskal–Wallis Rank Sum test. To test specifically for differences in pulse separations among behaviors, we filtered data to include only those abdominal pulses separated by <1 s, and repeated the aforementioned Kruskal–Wallis Rank Sum test. We then conducted pairwise Wilcoxon-Mann-Whitney tests comparing pulse separations across behavior groups. We applied a Holm correction for multiple testing. Since individual bees were monitored for different durations while performing behaviors inside cells, some bees could have disproportionately influenced the separation interval of the behavioral category to which they belonged. To account for any effect of individual bees on the timing between abdominal pulses, we performed a linear mixed effects logistic regression analysis using the R library lme4 (in package lmer test), with bee ID as random factor and behavior as fixed effect. Because our residuals were initially not normally distributed, we performed a rank transformation before conducting the regression analysis.

Thermal measures

We measured Tth and Tsurr across three timepoints (beginning, middle and end of each bee’s behavior duration). If we were to use all temperatures in our analyses, then we would have three Tth and nine Tsurr per bee. If we were to use only the middle temperature measurements (which might avoid behaviorally transitional complications), then we would have one Tth and three Tsurr per bee. To determine whether either method would affect behavior mean Tth or Tdiff, we compared mean Tth, then mean Tdiff of each behavior between the two methods using four Wilcoxon-Mann-Whitney tests. Corrections for multiple testing were not necessary. Heating behavior was confirmed by comparing a bee’s thoracic temperature to that of the surrounding region. For temperature difference analyses, because the data were not normally distributed and the sample size was relatively small, we applied the Wilcoxon-Mann-Whitney test with Holm correction. To see if the temperature associated with behavior differed across behaviors, we applied a Kruskal–Wallis Chi-square test using the Tth and Tsurr from the middle timepoint thermal measurements. We then conducted post-hoc pairwise Wilcoxon-Mann-Whitney tests with Holm correction on each of six combinations of behavior pairs. We repeated these methods for Tdiff. To determine whether Tdiff changed over the duration of a bee’s behavior in a cell, we used the R package nparLD (for nonparametric longitudinal data; Noguchi et al., 2012) to conduct a non-parametric ANOVA-type test. We applied the formula F1-LD-F1, which tests for group (behavior) differences, change over time, and the interaction between group and time. Tdiff was compared across temperature measurement periods 1, 2 and 3 for all bees, grouped by behavior. Before analyzing any thermal data, we corrected for the thermal signature of the thin film of giftwrap that functioned in enclosing the observation colony. To do this, we used thermographic measurements of the same neutral surface with and without the giftwrap film covering at a range of room temperatures from 26.5–43.6 °C. Differences between the two measurements at the same nominal temperatures were used to generate a set of correction values, which were then added as offsets to all thermal measurements of the colony behind the giftwrap (Dataset S5).

Limited-visibility test

To calculate the reliability of identifying behavior based on observing only the posterior tip of a bee’s abdomen, we applied a binomial test (Binomial Test Calculator, https://www.socscistatistics.com/tests/binomial/default2.aspx), with the null hypothesis that determination of behavior is random and not related to the actual correct behaviors. We then corrected for multiple testing using the Holm method. We set alpha at 0.05, report two-tailed P-values for all tests, and report errors as standard deviations. M.K.B. performed all statistical tests using R (R Core Team, 2019), except for binomial test on limited-visibility experiment.

Results

We conducted 49 surveys (21 nighttime, 28 daytime) of behaviors exhibited inside comb cells across 34.5 h. Absolute counts of each behavior differed across the surveys (Kruskal–Wallis rank sum test χ2 = 123.3, df = 3, P < 2.2 × 10−16). Of the 455 behavioral events monitored inside cells, bees spent 16.9% sleeping (n = 63), 76.4% maintaining cells (n = 362), 0.4% eating (n = 2) and 6.4% heating (n = 28). Bees slept for bouts of 1316 ± 1038 s (range: 257–3,346 s), maintained cells for 237 ± 257 s (range: 61–845 s), ate for 447 ± 233 s (range: 197–659 s), and heated for 956 ± 509 s (range: 452–2,214 s) (n = 7, 9, 3, 10 recordings of entire duration in cell for each behavior category, respectively). Behaviors were exhibited day and night, with no evidence of day-night bias for any behavior (2-sample Kolmogorov–Smirnov tests, D = 0.13, 0.26, 0.09; P = 0.99, 0.40, 1.00 for sleeping, maintaining cells and heating, respectively; eating sample size was too low for test to be meaningful; Fig. 5). None of the discontinuously ventilating bees exhibited visible signs of wakeful activity (larger movements of body, antennal movement, chewing, etc.), and because discontinuous ventilation covaries with other sleep signs (see above), “sleeping” is used as a shorthand for discontinuous ventilation + relative immobility inside cells, below.

Figure 5 Number of observations spent sleeping, maintaining cells, eating, or heating inside cells across time.

Note that number of observations during any given survey ranged from 1 to 17 bees. Sleeping data are presented top-down so comparisons of number of occurrences can more easily be compared within this behavior across time. Two eating occurrences are presented here, but a third eating event was recorded for ventilatory and thermal data between surveys (Figs. 6 and 8).

Figure 6 Ventilatory pulses of abdomen over time by worker bees that were sleeping, maintaining cells, eating, or heating.

Gray areas to the right of each pulse sequence signify post-recording periods (no data). Bees (y-axis) included 12 uniquely marked individuals (total n = 37 bees; 12 sleeping, 10 cleaning, three eating, 12 heating).

Ventilatory signatures as indicators of behavior

Ventilatory patterns differed among behaviors exhibited inside cells, as evident when plotting abdominal pulses (Fig. 6), and time between pulses (Fig. 7) (Kruskal–Wallis rank sum test, χ2 = 185.2, df = 3, P = 2.2 × 10−16). Discontinuous ventilation associated with sleep was identified by having discrete bouts of abdominal pulses, with the bouts separated by at least 10 s. Bouts of pulses were separated by 34.5 s ± 12.7 s (range: 10.1–336.6 s, n = 179 bout separations with a mean of 10 bout separations per bee across 12 bees). Pulses within bouts were separated by 0.27 s ± 0.06 s, when excluding pulse separations >1 s, which helped to exclude possible spontaneous abdominal jerks that appeared distinct from bouts of pulses (n = 1166 pulse separations with a mean of 97 pulse separations per bee across 12 bees).

Figure 7 Time between pulses of the abdomen (= pulse separations) when exhibiting different behaviors.

Ventilatory behavior is commonly distinguished as either continuous or discontinuous, depending on pattern of pulse separations. (A–D) Histograms displaying frequency of pulse separations when bees were sleeping, maintaining cells, eating, or heating. Y-axes break at frequency = 14 to show spread of data along x-axes; maximum y-values (1,166, 335, 328, 5,039) are superimposed above 14 in each plot. Pulse separations that were (E) long (>10 s) and typically associated with discontinuous ventilation, or (F) short (<10 s), by behavior. Pulses separated by <1 s are typical within pulse bouts that are separated by long pauses during discontinuous ventilation (associated with sleep), and are common throughout continuous ventilation. Note different y-axis scales in E and F. (npulse separations = 1,342 during sleeping, 490 maintaining cells, 394 eating and 5,525 heating; nbees analyzed = 12 sleeping, 10 maintaining cells, three eating, 12 heating).

Continuous ventilation (by bees maintaining cells, eating, or heating) rarely included separation of pulses by greater than 10 s (n = 28 out of 490 pulse separations when maintaining cells, 14 out of 394 when eating, and only 16 out of 5,525 when heating; Fig. 6), and instead featured relatively continuous abdominal pulses, which were separated by the same amount of time as sleeping bees, above (0.33 s ± 0.08 s for pulse separations <1 s, n = 5701 pulse separations with a mean of 328 pulses and a mean of 78 pulse separations per bee across 25 bees; linear mixed model after rank transformation, F3,34 = 1.19; P = 0.33).

Abdominal pulses can be difficult to discern when bees are very active (maintaining cells or eating) in cells, so ventilatory rates should be viewed in the context of whether or not a bee is exhibiting larger body motions.

Thermal measures as indicators of behavior

Body temperatures (Tth) differed from surrounding temperatures (Tsurr) (Kruskal–Wallis χ2 = 21.2, df = 3, P = 9.5 × 10−5), but only when bees were heating. Tth did not differ from Tsurr when bees were sleeping, maintaining cells, or eating (Tdiff = 0.20 ± 0.23 °C when sleeping, 0.20, ± 0.33 °C when maintaining cells, and 0.86 ± 0.31 °C when eating; n = 8, 10, 3 bees and W = 37.5, 58.5, 7, respectively; corrected P-values using Holm method = 1.0 in each case) (Fig. 8). Tth was only statistically different from Tsurr in heating bees (Tdiff = 2.62 ± 1.37 °C; n = 11 bees, W = 112, P = 0.0032). Heating bees’ Tdiff was greater than other bees’ Tdiff (vs. sleeping: W = 88, P = 0.0002; vs. maintaining cells: W = 110, P = 0.0006; vs. eating: W = 32, P = 0.044). Tdiff did not differ between sleeping and maintaining cells (W = 39, P = 0.96), nor did Tdiff differ between maintaining cells and eating (W = 2, P = 0.068), but eating bees’ Tdiff was greater than sleeping bees’ (W = 0, P = 0.044). A heating bee’s body temperature visibly differed from her surrounding temperature when using thermal imagery (Figs. 8, 9A and 9B; Movies S9 and S10), and we used this visible difference to initially identify heating bees, prior to analyzing ventilatory rates. Our aim here is to quantitatively confirm this difference (Tdiff) so that we can confidently associate heaters’ telltale heat emission with complementary behaviors (immobility + continuous ventilation) to confirm that the complementary behaviors alone can be used to distinguish heating bees from bees exhibiting the other behaviors. Ventilatory rates are important because a heating bee’s thoracic temperature fluctuates over time (Kleinhenz et al., 2003; Fig. 9C; Movie S11), and a relatively hot thorax does not necessarily mean a worker is actively performing as a heater, but could instead be transitioning into another behavioral state (Fig. 9D; Movie S12).

Figure 8 Temperatures of bees and of bees relative to their surroundings when exhibiting different behaviors.

Individual bees’ thorax temperatures relative to surrounding temperatures (Tdiff = Tth − Tsurr, left axis; colored circles), and absolute temperature of thorax (right axis; + signs) when sleeping, maintaining cells, eating, or heating. Solid lines represent mean temperature of Tdiff across bees for each behavior, and dashed lines represent mean temperature of thorax (Tth) across bees for each behavior. Temperatures are all surface measurements taken remotely with a thermal camera. Bees (x-axis) included 11 uniquely marked individuals (total n = 32 bees; eight sleeping, 10 cleaning, three eating, 11 heating).

Figure 9 Still images from thermal imaging videos.

(A) A worker heating while inside a cell, head facing left and part of abdomen, wings and hind leg extending outside cell, to the right. Image taken from Movie S9. (B) The two brightest spots, at left, each show the relatively hot thorax of a heating bee inside a cell, one immediately behind the plastic window of the hive (top arrow), and one that is one cell deep, seen through the wax wall of a cell (bottom arrow). These workers spent a total of ca. 5 min and 25 min heating inside these cells, respectively. Image taken from Movie S10 at 35 s after 21:41 h. (C) The arrows point to a heating bee with fluctuating body temperature. Images taken from Movie S11 at 10 s after 07:58 h and 2 min 48 s later. Total heating time exceeded 11 min. (D) Each bright spot is a relatively hot thorax, but of a bee maintaining cells, not heating. Image taken from Movie S12.

Time spent exhibiting a behavior inside cells (beginning, middle and end of stay) did not affect relative body temperature (Tdiff; ANOVA-Type statistic = 0.48, df = 1.7, P = 0.58).

Reliability of observing posterior tip of abdomen for identifying behaviors

We tested how reliable watching only the posterior tip of a bee’s abdomen is for identifying a behavior when a bee is inside a cell. Fifty-four human subjects correctly identified when honey bee workers were sleeping 86.6% of the time (n = 461 of 540 observations of 11 bees; binomial test, expected = 0.25; z = 32.3, P = 4.0 × 10−5; 13.5% misidentifications, with 8.1% identified as heating), maintaining cells 50.1% (n = 174 of 353 observations of seven bees; z = 10.5, P = 4.0 × 10−5; most common misidentification: 49.2% eating), eating 70.4% (n = 76 of 108 observations of two bees; z = 10.8, P = 4.0 × 10−5; most common misidentification: 31.5% maintaining cells), and heating 73.0% (n = 446 of 617 observations of 12 bees; z = 27.1, P = 4.0 × 10−5; most common misidentification: 18.5% maintaining cells). Participants typically reported difficulty determining behavior due to blurriness of abdomen (one video) or jostling of bee by other bees (1–2 videos). All percentages are means of percentages across bees to address effect of bee, some behaviors of which were more difficult than others to identify. For this reason, percentages may not sum perfectly to 100.

Discussion

Of the behaviors we recorded inside comb cells, sleep made up 16.9% of the observations, second only to maintaining cells. Maintaining cells and eating were easily identified when observing movement of body or mouthparts, and contents of the cell. Sleeping and heating bees lacked large movements of body or head, and were distinguished from each other using ventilatory rates (discontinuous vs. continuous pulses of the abdomen, respectively) and body surface temperature (relative to surrounding surface temperature). When visibility was restricted (i.e., when the contents of cells were obscured and only the posterior tips of honey bees’ abdomens were visible), maintaining cells and eating were difficult to distinguish from each other, but sleeping and heating were identifiable based on ventilatory rates and lack of major body motions alone (86.6% and 73.0% of observations were correctly identified, respectively). We used these two indicators to initially identify sleeping bees and, despite the relative ease of using thermography to distinguish heating bees, the same two indicators (ventilatory rate and lack of major body motions) appear most reliable to identify heating as well. We base this on the fact that a heating bee’s temperature can fluctuate, or confusion can arise when bees transition from one behavior to the next (Figs. 9C and 9D; Movies S11 and S12). We also base this on the high reliability of identifying heating bees in our limited-visibility reliability test, which we expect would increase by training observers for longer than 20–30 min.

This study’s findings match or differ from other studies in revealing ways. Foragers sleep more during the night than during the day (Kaiser, 1988; Sauer et al., 2003; Sauer, Herrmann & Kaiser, 2004; Eban-Rothschild & Bloch, 2008; Klein et al., 2008), but in this study sleep did not occur more at night (Fig. 5), suggesting that we were likely observing younger workers (e.g., cell cleaners and nurse bees). These younger “hive” bees sleep primarily in cells, and behave arrhythmically (Sauer, Menna-Barreto & Kaiser, 1998; Sauer et al., 1999; Eban-Rothschild & Bloch, 2008; Klein et al., 2008, 2014; but see a report of day-night differences inside cells in Moore et al. (1998)). Bees were sleeping in 16.9% of observations, which falls within the wide range of caste-dependent sleep observed inside cells by Klein et al. (2008) (1.6% observations of foragers—39.4% of cell cleaners). Comparisons with Lindauer (1952) are not feasible because he recorded data from only two individuals under relatively normal conditions, did not distinguish discontinuously ventilating or restful states from superficially similar behavioral states, and did not specify whether calculations were based on idleness exhibited within versus outside cells. Sleeping bees’ surface temperatures did not differ from their surroundings, and were slightly higher (Tth = 34.7 ± 0.8 °C, n = 8 bees) than were reported in “resting” bees by Kleinhenz et al. (2003), which were also measured inside cells (32.7 ± 0.1 °C–33.4 ± 0.3 °C, n = 5 bees). These resting bees exhibited discontinuous ventilation, with inter-bout durations lasting up to 58 s (vs. 34.5 s ± 12.7 s, lasting up to 337 s in this study). Heating bees are typically notably hotter than their surroundings, but the contrast was not as extreme in this study (Tdiff = 2.6 ± 1.4 °C; n = 11 bees) as it was in Kleinhenz et al. (2003) (4.2 ± 1.6 °C; n = 8 bees), but the body temperatures were equally high in both studies (Tth = 38.7 ± 1.6 °C, n = 11 bees; 38.3 ± 1.6 °C, n = 8 bees).

Limitations of study

Our observation hive approximated natural conditions in that it was kept in a relatively dark and warm room, featured combs spaced natural distances apart, contained food and brood, the queen was free-roaming, and an entrance allowed full access outdoors. Despite these similarities to natural nests, we supplied the colony with food ad libitum, and comb was limited to narrow slices attached on one side to a plastic window. Reports by Gontarski and Geschke (as communicated by Von Frisch (1967), p. 7), suggest that 500 or 500–1,000 members are sufficient for developing the same division of labor as in normal colonies, but we cannot know if our tiny colony (800–1,000 bees) developed a natural division of labor during this short study. It is important to note that the cells visible along one edge of each comb from which we collected our data may present behavioral biases, which would affect results related to the proportions of behaviors exhibited in cells reported in our surveys. Contents removed from edge cells to increase visibility of comb cells could have caused increased cell cleaning and building activity. We wanted to increase the likelihood of observing sleep in the visible cells, so our emptying of edge cells could have caused a higher rate of discontinuous ventilation within these edge cells. Small numbers of brood or small size of comb could have resulted in unnatural rates of heating, as well. Our limited-visibility reliability test for predicting behaviors featured a lateral view of abdominal tips (Fig. 4) when the typical view would be posterior view of abdominal tips (Fig. 1A). The limited-visibility test included only three bees eating, and the sole training video devoted to eating did not include eating behavior, only presence of food with description of behavior.

Why sleep inside cells?

Accounting for sleep inside honeycomb may help to resolve contradictory or confusing evidence reported under less natural conditions (Sauer, Menna-Barreto & Kaiser, 1998; Eban-Rothschild & Bloch, 2008). If we can rely on discontinuous ventilation + absence of major body motions as markers of sleep in limited-visibility, in-cell situations, the youngest adult workers spend more time asleep than later in life (Klein et al., 2008). Sleeping more earlier in life is normal across animals, and much research has considered the current utility of this standard feature of sleep ontogeny. Cell cleaners and nurse bees sleep primarily inside cells that are located in or close to brood comb, and this could be for a variety of functionally interesting reasons (see Klein et al., 2008). Comb cells may protect sleeping adults from being disturbed, which could reduce the damaging effects associated with sleep fragmentation. Comb cells could provide warmth for regenerative or cognitive processes, or serve as a site that reduces sleepers’ interference with other workers bustling about the comb. Alternatively, sleeping in cells could be a nonadaptive behavior, during which honey bees simply use comb cells as a default site between acts of cell maintenance, nursing, or heating.

Poets, philosophers and scientists have long pondered the societal marvels of honey bee colonies (Preston, 2006), and making visible the bees’ activities is a pursuit that has changed our understanding of what nonhuman animals are capable of. Activities, like sleeping, can be difficult to access, particularly when performed inside honeycomb hidden within a dark tree hollow. What specific benefits are conferred by bees sleeping inside cells awaits further investigation, and likely will depend on technical innovations involving noninvasive imaging of standard hives or natural nests, or testing sleep and sleep loss in noncircadian subjects.

Conclusions

The best view of a honey bee inside a honeycomb cell is typically restricted to the tip of its abdomen, under the best of circumstances. We hypothesized that even with such constraints, the capacity to identify sleep and other behaviors can be high, based on brief observations of the ventilatory rates (discernible by timing of abdominal pulsing motions) combined with the presence or absence of major body movements. Viewing bees inside cells using a special hive and filming with an infrared-sensitive camera and thermal camera made identifying all behaviors relatively easy in this study, but identifying sleeping or heating bees was also reliable with the limited visibility available to an observer without this special hive or thermal camera. Simply observing ventilatory movements, as well as larger motions evident in the tip of a bee’s abdomen was sufficient to noninvasively identify sleeping or heating inside comb cells. Cell maintenance was frequently confused with eating under limited visibility conditions, but both were clearly distinguishable from sleeping and heating. Sleeping and heating were accurately identified (86.6% and 73.0% of observations, respectively) by observing ventilatory rates (discontinuous versus continuous, respectively), combined with a lack of major body movements. Although reliability of identifying behaviors was high, the specialized hive we used may have biased proportions of time bees slept, heated, ate, or maintained cells. Sleep appeared frequently enough to suggest that it is an important behavior experienced within honeycomb cells, supporting previous examinations of sleep inside comb cells, and lending credibility to future ventures, which can rely on similarly less invasive manipulations to reveal the dynamics and functions related to sleep in nature.

Supplemental Information

Supplemental Information 1 Supplemental Videos.

Click here for additional data file.

Supplemental Information 2 Infrared-sensitive video of bee sleeping inside cell.

Bee, center, is facing left with venter facing up. Note worker bees maintaining (cleaning or building) other cells.

Click here for additional data file.

Supplemental Information 3 Infrared-sensitive video of eating inside cell, with mouthparts extended and body less fully inserted in cell.

Bee is facing left with dorsum facing up. Note worker bees maintaining (cleaning or building) other cells.

Click here for additional data file.

Supplemental Information 4 Infrared-sensitive video of heating inside cell.

Bee, lower left, is facing left with venter facing observer (sideways).

Click here for additional data file.

Supplemental Information 5 Infrared-sensitive video of sleeping and heating inside cells.

Sleeping bee, center, is facing left with dorsum facing up, and is to be compared with heating bee, at right, facing right with dorsum facing observer (sideways).

Click here for additional data file.

Supplemental Information 6 Infrared-sensitive video of worker bee inside cell.

Gray box obscures cell innards, and small light gray rectangle marks bee of interest. This was one of 30 modified video clips used to test reliability of identifying inside-cell behavior from what is visible outside the cell. (Behavior? Answer: sleeping)

Click here for additional data file.

Supplemental Information 7 Infrared-sensitive video of worker bee inside cell.

Gray box obscures cell innards, and small light gray rectangle marks bee of interest. This was one of 30 modified video clips used to test reliability of identifying inside-cell behavior from what is visible outside the cell. (Behavior? Answer: heating)

Click here for additional data file.

Supplemental Information 8 Infrared-sensitive video of worker bee inside cell.

Gray box obscures cell innards, and small light gray rectangle marks bee of interest. This was one of 30 modified video clips used to test reliability of identifying inside-cell behavior from what is visible outside the cell. (Behavior? Answer: eating)

Click here for additional data file.

Supplemental Information 9 Infrared-sensitive video of worker bee inside cell.

Gray box obscures cell innards, and small light gray rectangle marks bee of interest. This was one of 30 modified video clips used to test reliability of identifying inside-cell behavior from what is visible outside the cell. (Behavior? Answer: cleaning)

Click here for additional data file.

Supplemental Information 10 Thermal imaging video of heating bee inside cell.

The thorax is relatively hot, the abdomen is continuously ventilating, but the bee is otherwise immobile. Video plays close to actual time (30 images per second).

Click here for additional data file.

Supplemental Information 11 Thermal imaging video featuring many acts of heating and cell maintenance (cleaning or building) inside cells.

Video captured 1 image per second.

Click here for additional data file.

Supplemental Information 12 Thermal imaging video featuring many acts of heating and cell maintenance (cleaning or building) inside cells.

Heaters’ thoracic temperatures fluctuate over time (e.g., bee featured in Fig. 9C is from this video at 10 s after 07:58 h and 2 min 48 s later). Video captured three images per second.

Click here for additional data file.

Supplemental Information 13 Thermal imaging video of workers maintaining (cleaning or building) cells.

Each bright spot is a relatively hot thorax, but of a bee maintaining, not heating, cells. This video plays close to actual time (30 images per second), allowing the viewer to observe a behavior closely for what it is.

Click here for additional data file.

Supplemental Information 14 R analyses and visualizations.

This set of R scripts conducts all of the visualization and statistical analyses for the “Slumber in a cell” project.

Click here for additional data file.

Supplemental Information 15 Dates and times for the surveys performed on all bees exhibiting any of the four behaviors while inside cells.

The behaviors are expressed as totals for each behavior, and as proportions of the total.

Click here for additional data file.

Supplemental Information 16 Identical to the Dataset_S1.csv spreadsheet, except that it includes the column total.beh.

This column is necessary for executing the statistics script related to the behavior surveys, which needs the totals in long format to conduct a Kruskal–Wallis test.

Click here for additional data file.

Supplemental Information 17 Worksheet including times for each abdominal pulse for each of the monitored bees, as well as the calculated separations between each pulse.

The columns that end with “by3” are simply the corresponding columns multiplied by 0.3 because the original videos were slowed to 0.3 speed to facilitate observing and marking the abdominal pulses. Results were later restored to the original timestamps. The column “LBB” stands for “Look Between Bouts”, and it excludes any pulse isolated by >5 s. “Event” = “pulse”, as defined in the paper. All “event” (pulse) times and “event” (pulse) separations are measured in milliseconds.

Click here for additional data file.

Supplemental Information 18 Worksheet containing IDs, behaviors, and surface temperatures of the thorax and surroundings for each bee.

The time announced and time beginning columns are used for cross-referencing bee identifications with other records. Mean Tth and Tsurr were calculated in the indicated columns using both methods of calculation: (1) T2 only, and (2) T1, T2 and T3 averaged. Delta temperatures (differences between Tth and Tsurr) are also indicated. Standard deviations are included for all temperature measurements and means. Below the raw data, we have calculated mean, standard deviation, minimum and maximum durations for each behavior.

Click here for additional data file.

Supplemental Information 19 Adjusted temperature measurements.

The giftwrap used as the observation hive’s window produced a nonlinear error when recording temperature as temperature increased, so we adjusted absolute temperature measurements. Adjusted temperatures are listed here. Data were collected by Christian Lutsch.

Click here for additional data file.

Jürgen Tautz, Hartmut Vierle, and the entire BEEgroup generously hosted B.A.K.’s research at the University of Würzburg. Sven Mayer and Dirk Ahrens-Lagast ably and graciously assisted with innovative beekeeping. Markus Mika helped to translate a portion of Lindauer’s (1952) manuscript for us. Alistair Alesch edited video clips for testing predictability of behaviors based on visibility of abdominal tips alone, and University of Wisconsin—La Crosse students took the limited-visibility test. David Reineke (University of Wisconsin—La Crosse) and Jeff Oliver and Keaton Wilson (University of Arizona Library) contributed statistical and R coding consultation. Logan Schoolcraft provided help in data preparation. Bryant Tran assisted with video editing and file processing. Laura Still and Melody Latronico assisted with some preliminary data transcription. Two anonymous reviewers and Stephen Pratt provided profoundly helpful critiques and suggestions. As always, we thank our sleeping, cell-maintaining, eating and heating honey bees.

Additional Information and Declarations

Competing Interests

Author Contributions

Field Study Permissions

Data Availability

The authors declare that they have no competing interests.

Barrett A. Klein conceived and designed the experiments, performed the experiments, analyzed the data, prepared figures and/or tables, authored or reviewed drafts of the paper, and approved the final draft.

M. Kathryn Busby analyzed the data, prepared figures and/or tables, authored or reviewed drafts of the paper, and approved the final draft.

The following information was supplied relating to field study approvals (i.e., approving body and any reference numbers):

Honey bee research was conducted using bees from an on-campus bee yard with approval by the University of Würzburg (Würzburg, Germany).

The following information was supplied regarding data availability:

Data are available in the Supplemental Files. All Supplemental Information, plus additional files, are also available through Open Science Framework: Klein, Barrett A., and M.K. Busby. 2020. “Slumber in a cell: honeycomb used by honey bees for food, brood, heating… and sleeping.” OSF. July 3. osf.io/eux5y.

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
