# Peer review of "Slumber in a cell: honeycomb used by honey bees for food, brood, heating… and sleeping"

_PeerJ, doi:10.7717/peerj.9583_

## Round 0.1 · original submission · Major Revisions

Dear Drs. Klein and Busby:

Thanks for submitting your manuscript to PeerJ. I have now received three independent reviews of your work, and as you will see, the reviewers raised some concerns about the research. Despite this, these reviewers are optimistic about your work and the potential impact it will have on research studying honey bees. Thus, I encourage you to revise your manuscript, accordingly, taking into account all of the concerns raised by all three reviewers.

Please improve the presentation, clarity and organization of your manuscript (many suggestions raised by the reviewers). Please also work to make the separate subjects flow cohesively. As is, the manuscript is rather disjointed.

Please also ensure that your figures and tables contain all of the information that is necessary to support your findings and observations. Revise incorrect information. The Materials and Methods appear to be missing important information. All statistical methods should be adequately described such that they are repeatable.

I look forward to seeing your revision, and thanks again for submitting your work to PeerJ.

Good luck with your revision,

-joe

·

Basic reporting

This study categorized the behavior of bees inside comb cells by using a specially designed observation hive that gave a full view of the cell interior. The central findings are that bees often sleep in comb cells, and that this behavior can be reliably detected by inspecting the overall motion and ventilation pattern of the rear end of the bee, which is readily observable in a conventional hive. These findings are of interest to students of bee behavior and sleep, and the methods and data are sound, to the degree that I can evaluate them. However, this evaluation is limited, because the organization and presentation of the work make it difficult to follow. I suggest a major reorganization of the text to better explain the study.

The introduction should start with the central topic of the study, sleep in honeybees, rather than a discussion of comb and hive design. Instead, sleep and the behavior of bees in cells is only introduced in the second paragraph.

The Materials and Methods section needs a plain statement, early on, of the basic design of the experiment. My take on the design, which may be inaccurate in places, is as follows. A small colony of bees was set up in an observation hive that permitted an interior view of some cells. Over a period of two days, the hive was video recorded and simultaneously imaged with a thermal camera. The videos were then scanned and the behavior of bees in the visible cells was classified into one of four categories, based on a variety of criteria. These data were taken as an accurate description of the bees’ behavior. Then, naïve viewers looked at a subset of the videos, which had been masked so that only the posteriors of the bees were visible, as in a normal hive. They classified the bees into the same four categories, and their performance was compared to the accurate ground truth from the interior views.

There are a number of methodological points that I could not determine from the text. Why were some bees marked? Were only marked bees subjected to behavioral categorization? How long was the hive recorded? How were bees selected for behavioral categorization? Was every visible bee observed, or only a subset? Was the hive repeatedly scanned, or were focal areas watched continuously? Were the behaviors observed and classified only after the fact, from videos, or were they watched live and the classifications recorded? The latter is suggested by a reference to “announcing a bee’s behavior” (line 167), but this is not explicitly described anywhere. In short, there needs to be a straightforward and complete description of how the study was conducted.

The section on behaviors of interest would be clearer if it systematically went through the four behavioral categories and explained exactly what observations led to putting a bee in each category.

The statistical analysis is a bit hard to follow, because the nature of the sampling regime is not clear, as noted above. What is the meaning of the “surveys” mentioned at line 240? What exactly is an event? Over what sample were mean percentages of bees in each behavior calculated?

Other points:
Line 181: What is JWatcher and how was it used to record ventilatory rates?
Line 254: Why were only 11 bees examined for ventilatory signatures of sleep?
Line 261: What is an event separation?
Figure 7b: What is the difference between the upper and lower plots? What exactly are the events whose separations are plotted? The y-axis would be a bit clearer scaled in seconds, rather than milliseconds.

Experimental design

The description of how the temperature of a bee’s surroundings was measured is not clear. Wouldn’t a circle with a radius of one bee length, centered on the bee’s thorax, include the entire body of the bee? This would confound her temperature with that of her surroundings.

Lines 268-285: This analysis of temperature differences among behavioral categories is circular, is it not? Temperature was the means by which bees were classified as heaters, so it is not informative to then do a statistical test showing that their temperature is different from the others.

Validity of the findings

No comment

Additional comments

No comment

Reviewer 2 ·

Basic reporting

General: The authors used both thermal imaging and detailed video analyses to observe the behavior o bees concealed inside their comb cells, they used vertical comb setting (Lindauer hive) to see what is happening inside the cells. This enabled to compare how the bee behavior is seen in normal view, which reveals only the tip of the bee abdomen, with the actual behavior of the bee inside the cell. The finding show that the frequency of abdominal ventilation movements seen from outside can be used to identify the bee behavior inside the cell. Heating brood cells or sleeping were identified with high level of certainty. The identification of feeding and cleaning behaviors was less accurate without seeing the cell contents. The study shows valuable, unique and beautiful videos and thermal imaging of the different behaviors of honey bees inside cells. The findings of this study will help future investigations of bees’ behavior inside cells, specifically sleep. The Introduction is interesting, relevant and well-written. The hypothesis is well formulated and explained.
Figures and tables:
1. The resolution of the figures is too low so impossible to zoom in, specifically on the pulses in fig. 6.
2. The figures showing still images from the movies are redundant, given the movies. The figure annotations can be adapted and added to the beautiful movies, which lack legends and annotations.
3. Fig. 5: the ticks on the Time axis are distributed equally, but the time gap between the ticks varies.
4. Fig. 7: the legends are not consistently phrased: sometimes they describe the findings (here) and sometimes describe the plot details (e.g., fig. 6, 8). A: graph should have more ticks and better resolution to demonstrate the 10sc threshold. Please explain the plots in (B) in detail.
5. Table 1: please increase the spaces between the table lines
Discussion: There are some redundant repetitions on the data shown in Results section; e.g., L. 301, L.309, L.323 etc.

Experimental design

The methodology is not sufficiently explained and some statistics is missing:
Major:
Materials and Methods
1. The description of the experimental design is not sufficiently well organized. The descriptions of behavioral recordings, thermal measurements and abdominal pulse measurements are mixed. Only after looking at the data sheets, it becomes clearer that these are 3 sets of recordings (is that correct?).
2. It is ambiguous whether thermal measurements were used to validate every behavioral bout (e.g., L. 170-171, L. 277-278).
3. The methodology for scoring behaviors is unclear- was the hole hive scan sampled?
Results:
4. L. 240-244: If the data was obtained by scan sampling, one must make sure that sampling time was equal between day and night. The duration of the behavioral recordings is not reported. According to Fig. 5, recording times varied (6-20 minutes). I suggest to correct the analysis such that day/night sampling duration is equal to prevent bias.
5. L.250-251: there are no statistics backing the claim that ventilatory patterns differed. One could use contingency table comparing event separations above and below 10s between the behaviors.
6. 268-270: the claim is not supported with statistics. The authors could test if the Tdiff is significantly different from zero in each behavior, although the sample sizes are extremely low. In anyway, claims it should be backed with statistics.
Minor:
Materials and Methods
1. Room temperature is not reported- may affect the frequency of heating behavior, for example.
2. How were observations assigned to night and day? According to sunrise/set times?
3. The age of the focal bees is not reported. This can affect the distribution of observed behaviors because of the age related division of labor in honey bees.
4. L. 164-165: claim there is no data on brood tending, but ‘feeding’ definitely qualifies as brood tending.
5. L.195: to which analyses do the authors refer? Only after comparing the sample sizes, it is possible to understand that it is the abdominal pulses’ analysis.
6. 214-216: results of this analysis are absent.
7. The terms ‘abdominal pumping’ and ‘abdominal pulses’ are used alternately, causing confusion. Please use a consistent term.
Results
8. 270-271: can you please clarify what does the claim “Tdiff was only statistically significant in heating bees…” mean? Tdiff significantly differed from zero?
9. 272-275: correction for multiple comparisons is required.
10. 294-296: the analysis is not clear: what is reported? Is it the percentage of the most common mistake for each behavior? If so, why should the percentages sum to ~100%?

Validity of the findings

Major:
1. Sample sizes for the thermal measurements and for abdominal ventilation measurements were extremely small- with just 2-9 or 2-11 events per behavioral category, respectively (fig. 6 & 8),. There were no repetitions on the experiment with more colonies. It is hard to reach concrete conclusions based on this data set. For example, the videos show that cleaning bees have high thorax temperature but the analysis shows no difference between cleaning and sleeping bees, is this conclusion valid considering the small sample size? Feeding behavior was characterized based only on 2 behavior bouts. This must be addressed.
2. The only criteria for distinguishing sleep and heating is abdominal ventilation frequency. Therefore, showing differences in that measure between the 2 behaviors is slightly tautological. Please address this point.
3. 216-217: This is not clear: why compare the distributions? One would compare the frequency of each behavior during night and day (using binomial test for example), assuming that the sampling duration was even during night and day.
4. L.319-325: the conclusion that “sleeping was not circadian” does not follow from the data. This study did not monitor the circadian rhythms of individual bees; it monitored what happens inside given cells. Individual bees may still have a circadian rhythm. For the same reason, the percentage of sleep occurrence in cells cannot be compared with the time an individual spends sleeping.
5. L.342-352: I agree with the authors on the limitation- The significance of the proportions reported for each behavior is questionable. I think the authors should provide explicit estimation on the direction of expected biases- for example, more empty cells/young bees result in higher frequency of sleep and cleaning. Can we really assume that the colony developed natural demography and division of labor after such short period of time (the experiment lasted only 4 days, L. 146)?

Reviewer 3 ·

Basic reporting

No comment

Experimental design

The methods are generally described well and in detail, but a few details should be added:

- Description of the overall organisation of the video/thermal image recordinging sessions over the two "data collection" days (23/24 August): what triggered the start/end/duration of a recording session, what were the rules for switching from the overall view of the entire hive to the close-up record of individual cells/bees and vice versa; how were long cell visits (longer than a continuous recording session) handled? Lines 165-166: were there also record periods when the author was NOT present and did not announce the behaviour?

- Use of the JWatcher software (line 181): please specify how this software was used to record the ventilatory pulses, e.g. via image analysis or via a human observer manually pressing a key when ventilation occurred, and what were the limititations of this method (shortest time between two pulses that could be recored)?

Validity of the findings

No comment

Additional comments

Other suggestions for improvement (not critical):

Line 294: It may be interesting to re-examine the 8.7% misidentification of heating/sleeping and the 18.5% misidentification of cleaning/heating to find out why they were not classified correctly by all persons. This may improve the authors' suggested proxy method and help avoiding/reducing wrong identifications in the future when using this method.

Table 1, category "cleaning": the authors should reconsider if the observed bees that were assigned to this category really performed only "cleaning" work or also construction work, i.e. massive manipulation of the wax comb: The appearance of cell walls in movies 1-4 suggest that some cell walls apparently have been firmly attached to the front pane by the worker bees. On the other hand, movies 1, 2 and 4 and also the thermographic videos of the entire hive show many of the exposed cells with their cell walls reduced to half their normal length/depth, or less. E.g., in movie 1 the bee in the centre of the image has fully entered a cell but its entire abdomen is outside the cell due to the reduced cell length. This implies that many "construction workers" must have been visiting these cells. Also in movie 1, the cell visitor in the bottom part of the right comb is clearly chewing pieces of wax off the cell wall and relocating them (e.g., during the first 25 s of this record, again on 8:32:57 PM, 8:33:04 PM, 8:33:31PM - 8:33:48PM of the original time in the image). Also the bee on the left side (third cell from bottom, 8:33:07 - 8:33:26 PM) is clearly chewing wax, not simply cleaning the cell.

Minor corrections (sorted by order of appearance in the text; no priorities):

Line 97: The citation of Flanigan 1973 seems to be missing "et al."

Lines 113-114: Kleinhenz et al., 2003, are cited correctly for observations of ventilation of bees inside cells. However, the mention of ventilatory pumping of bees outside cells should rather cite Bujok et al., 2002 (Bujok B, Kleinhenz M, Fuchs S, Tautz J: "Hot spots in the beehive", Naturwissenschaften 89: 299-301).

Line 279: the citation "Kleinhenz, 2003" seems to be missing "et al."

Lines 329-331: The authors claim that the publication of Kleinhenz et al. (2003) "did not report timing or durations of in-cell rest (or sleep)" although the observation time is stated and Table 7 in that publication provides durations of rest inside cells ranging from 4.8 to 14.5 min (5 resting bees).
Figure 4 (caption): According to the caption, the (correct) answers on the partially obscured behaviours (Movies 5-8) are included. However, I am unable to find these answers in the videos despite watching them until they end.

General comments/suggestions/clarifications (sorted by order of appearance in the text; no priorities):

Lines 115-117: The authors suggest that ventilatory activity may be a more suitable indicator of (honeybee) sleep under natural or close to natural conditions because the other indicator - antennal movement - is difficult or impossible to gauge in a nest. In this context, and to further support the authors' proposal, it may be worth mentioning that antennal immobility as indicator of sleeping bees may even be misleading: This is because brood-incubating bees on the comb surface, which may be mistaken as sleeping bees due to the absence of large body movements, also keep their antennae nearly immobile on the comb surface and only slow antenna movements may occur (Bujok et al. 2002, details see previous comment).

Lines 136-137, 350-352: the removal of cell contents to make the exposed cells accessible to cell visitors may present an additional bias (in this case: on cell cleaning behaviour) that is not discussed/mentioned in lines 350-352.

Lines 137-138: if available, it would be interesting to know the approximate numbers of these different cell types/contents on the comb pieces, and also the contents of those cells in the 2nd row (behind/adjacent to the exposed cells that were used to study cell visiting bees).

Lines 140-141: if available, an estimation of the error of thermographic temperature readings due to the colour marking, as determined in the preliminary tests, should be given.

Line 148: if available, the room temperature during the observation periods should be indicated. If standard temperature readings from inside the hive exist (e.g., air temperature measured via thermocouples or similar), those would also be interesting to know.

Line 149: Please add if there any thermal insulation covering the hive outside the recording periods

Line 167-168: Assuming that the color mark on the bees cannot be read from the greyscale infrared video nor from the thermographic video: how was the color mark identified (vocal record by the author when present during the recording?) Was the behaviour of unmarked bees also evaluated?

Line 178: The "surrounding" temperature was calculated as a mean from "the surface temperature of anything" "within a circle (in the thermographic image) with radius of one bee's body length from middle point of thorax", "including a portion of the focal bee herself". This sounds as if the focal bee was (nearly) completely within the measurement circle... An image should be provided to illustrate where exactly the "surrounding" temperature was measured and how much the focal bee itself (or even other bees in neighbouring cells, if those were visited at the same time?) contributed to this measurement. Also, a statement may be necessary to explain that the camera settings were adapted for the bee thorax, not for, e.g., wax.

Line 179-180: The measurement circle for the "surrounding" temperature includes "a portion of the focal bee"(see previous comment) but is used to calculate temperature differences between the focal bee and the "surrounding" temperatue. This implies a systemic mistake that may lead to smaller temperature differences between the bees and the surrounding. The size of this error depends on the proportion of the bee within the designated circle area (see comment on line 178) and on the bees' temperature. Assuming that bees at rest/sleep have a temperature at or near ambient, this mistake would be small. In case of bees with elevated temperature (in particular, if the thorax temperature is actively raised above ambient temperature and if the thorax lies within the circle for measurement of "surrounding temperature") the calculated temperature difference would be smaller than it actually is. This should be considered by the authors.

Line 203: please specify whether the headlamp was a cold light source or one using heat producing bulbs, and if any interference of the lamp with the thermographic measurements may be expected (e.g., reflection of thermal radiation from the lamp on the plastic film covering the hive).
Line 205: please provide the source for the given accuracy, e.g. general specification by the manufacturer (camera manual), calibration certificate before start of the work, or own comparative measurements on dead bees.

Line 210: please specify how the spoken information in the audio record (Digital Voice Recorder) was synchronised with the video records (e.g., by speaking the video time index before identifying a bee and its behaviour)?

Line 255 and 260: should it read as "for event separations < 10 s", or why is there another limit value ("< 1 s") to filter certain ventilation events? This limit has not been introduced before.
Line 311: it is not clear what "the average heating bee" is.

Figure 9a: there seems to be a lot of noise or wrong calibrated pixels in the close-up thermal image (and in movie 9), making it unlikely to meet the given accuracy of 1°C or 1% (line 205). Any explanation?

Supplementary material (movies): Data collection took place on 23/24 Augut but movie 3 dates 22 August and movie 10 and 12 date 20 August. Auxiliary still images and videos for demonstration purposes only (not used to collect data) should be marked as such.

---

## Round 0.2 · Minor Revisions

Dear Drs. Klein and Busby:

Thanks for revising your manuscript. The reviewers are very satisfied with your revision (as am I). Great! However, there are a few minor edits to make. Please address these ASAP so we may move towards acceptance of your work.

Best,

-joe

·

Basic reporting

The language and graphics are clear, correct, and thorough.

Experimental design

The research question is compelling and the study's design well-suited to answer it.

Validity of the findings

The conclusions are well-grounded in the data, and the authors are candid about limitations

Additional comments

The revised version of this manuscript is greatly improved and makes for a very interesting contribution about sleep in bees and about how it can be effectively studied. The experimental logic and methodology are now quite clear. I have only a few minor comments and suggestions, by line number.

177-179: Was the glass of the hive replaced with giftwrap only for the thermal recordings, or was it in place for the whole experiment? I would think that going back and forth would damage the comb attached to the glass.

351: What (roughly) were the magnitudes of the giftwrap correction factors?

427-439: The authors give a good measure for the sensitivity of their behavioral classifications: the proportion of true positives: e.g., cases of sleeping that were correctly identified as sleeping. For completeness, it would also be good to give a measure of specificity: i.e., the number of true negatives. For example, the authors could report 1 – p, where p is the proportion of all non-sleeping behaviors that were incorrectly identified as sleeping. They could do this for all four behaviors, or just for sleeping, since that is the main behavior of interest. If the measure I suggest seems a bit convoluted, they could also just report p.

Reviewer 2 ·

Basic reporting

The authors did a great job addressing all comments.

Experimental design

No further comments

Validity of the findings

No further comments

Additional comments

No further comments

---

## Round 0.3 · accepted · Accept

Dear Drs. Klein and Busby:

Thanks for revising your manuscript based on the concerns raised by the reviewers. I now believe that your manuscript is suitable for publication. Congratulations! I look forward to seeing this work in print, and I anticipate it being an important resource for groups studying honey bees. Thanks again for choosing PeerJ to publish such important work.

Best,

-joe